# Occurrence and multilocus genotyping of *Giardia duodenalis* in captive non-human primates from 12 zoos in China

Xueping Zhang[1☯], Liqin Wang[2☯], Xinting Lan[1☯], Jiaming Dan[1☯], Zhihua Ren[1], Suizhong Cao[1], Liuhong Shen[1], Junliang Deng[1], Zhicai Zuo[1], Shumin Yu[1], Ya Wang[1], Xiaoping Ma[1], Haifeng Liu[1], Ziyao Zhou[1], Yanchun Hu[1], Hualin Fu[1], Changliang He[1], Yi Geng[1], Xiaobin Gu[1], Guangneng Peng[1]*, Yufei Wang[3]*, Zhijun Zhong[1]*

**1** College of Veterinary Medicine, Key Laboratory of Animal Disease and Human Health of Sichuan, Sichuan Agricultural University, Chengdu, China, **2** The Chengdu Zoo, Institute of Wild Animals, Chengdu, China, **3** Department of laboratory medicine, The Third Medical Center, General Hospital of the Chinese People's Liberation Army, Beijing, China

☯ These authors contributed equally to this work.
* zhongzhijun488@126.com (ZZ); yufeiwang21@yahoo.com (YW); pgn.sicau@163.com (GP)

**Data Availability Statement:** All relevant data are within the manuscript and its Supporting Information files.

## Abstract

*Giardia duodenalis* is a common enteric protozoan that infects a range of hosts including humans and other mammals. Multilocus genotyping of *G. duodenalis* in captive non-human primates (NHPs) from zoos in China is limited. In this study, we evaluated 302 NHP fecal samples collected from 32 different NHP species. The primates were from 12 zoos distributed across eight provinces and two municipalities (Chongqing and Beijing) of China. The overall infection rate was 8.3% (25/302). The six *G. duodenalis*-positive zoos and their infection rates were: Suzhou Zoo (40.0%, 4/10), Yangzhou Zoo (22.2%, 2/9), Dalian Zoo (16.7%, 4/24), Chengdu Zoo (12.8%, 6/47), Guiyang Forest Wildlife Zoo (12.1%, 7/58), and Changsha Zoo (4.7%, 2/43). Molecular analysis of three loci, beta-giardin (*bg*), triose phosphate isomerase (*tpi*), and glutamate dehydrogenase (*gdh*), showed high genetic heterogeneity, and seven novel subtypes (BIII-1, MB10-1, WB8-1, B14-1, MB9-1, DN7-1, and BIV-1) were detected within assemblage B. Additional analysis revealed 12 different assemblage B multilocus genotypes (MLGs), one known MLG and 11 novel MLGs. Based on phylogenetic analysis, 12 assemblage B MLGs formed two main clades, MLG-SW (10–12, 18) and MLG-SW (13, 14, 16, 17), the other four MLG-SW (15, 19, 20, 21) were scattered throughout the phylogenetic tree in this study. Using multilocus genotyping, this study expands our understanding of the occurrence of *Giardia* infection and genetic variation in *Giardia* in captive non-human primates from zoos in China.

## Introduction

*Giardia duodenalis* is an intestinal parasite that causes giardiasis in humans and animals. *Giardia duodenalis* infection may be asymptomatic or elicit several clinical symptoms including diarrhea, vomiting, weight loss, abdominal cramps, and nutrient malabsorption [1, 2]. *Giardia duodenalis* commonly infects non-human primates (NHPs), and causes both veterinary and

**Funding:** This work was funded by the National Key Research and Development Program of China (2018YFD0500900, 2016YFD0501009) and the Chengdu Giant Panda Breeding Research Foundation (CPF2017-05, CPF2015-4).

**Competing interests:** The authors have declared that no competing interests exist.

public health problems [3–7]. In NHPs, giardiasis causes diarrhea and ill thrift, especially in young animals [8].

To date, there have been numerous studies about the *Giardia duodenalis* infection for non-human primates in the world, such as in Thailand (7.0%, 14/200) [9], Uganda (11.1%, 9/81) [10], India (31.2%, 53/170) [11], Netherlands/Belgium (61.6%, 159/258) [12], Italy (50.0%, 5/10) [13] and Spain (70.0%, 14/20) [14]. In China, *Giardia duodenalis* infection rates in ten zoos are reported between 0% to 44.0%, including Changsha Wild Animal Zoo (44.0%, 33/75), Gui-yang Zoo (30.0%, 15/50), Beijing Zoo (22.2%, 16/72), Shanghai Wild Animal Zoo (20.9%, 14/67), Taiyuan Zoo (13.6%, 9/66), Wuhan Zoo (7.6%, 5/66), Shijiazhuang Zoo (11.2%, 10/89), Shanghai Zoo (8.2%, 5/61), Bifengxia Zoo (0%, 0/24) and Chengdu Zoo (0%, 0/11) [8, 15]. *Giardia duodenalis* has at least eight assemblages (A-H), only assemblages A, B, and E have been detected in NHPs, with assemblage B dominating [8–13, 15–20]. Assemblages A and B, which are consider potentially zoonotic, were reported in NHPs from zoos [8–16], thus, NHPs may play a role in the transmission of *G. duodenalis* to humans.

To date, most Chinese studies evaluating *G. duodenalis* infection in NHPs have focused on a single zoo or localized area. Only three studies have extended their investigation to include a larger geographical region [8, 15, 16]. Ongoing epidemiological surveys on intestinal zoonotic parasites of *G. duodenalis*, expanded previous studies to large-scale investigation of zoos and NHP species in China. We used multilocus genotyping to evaluate 302 NHP fecal samples (including 32 primate species) from 12 zoos distributed across eight Chinese provinces and two municipalities (Chongqing and Beijing), to better understand *G. duodenalis* infection in captive NHPs throughout China.

## Materials and methods

### Ethics statement

This study was reviewed and approved by the Institutional Animal Care and Use Committee of Sichuan Agricultural University under permit number ZXP-2018303052. Prior to the collection of fecal specimens from NHPs, permission was obtained from the owners.

### Sample collection

Fecal samples from 302 NHPs (including 32 primate species) were collected from March 2018 to January 2019 (S1 Table). The samples from 12 zoos are distributed throughout China (Fig 1), including Beijing Zoo (n = 12), Chengdu Zoo (n = 47), Changsha Zoo (n = 43), Chongqing Zoo (n = 33), Dalian Zoo (n = 24), Guiyang Forest Wildlife Zoo (n = 58), Guangzhou Zoo (n = 8), Kunming Zoo (n = 16), Nanjing Zoo (n = 16), Shaanxi Rare and Wildlife Zoo (n = 26), Suzhou Zoo (n = 10), and Yangzhou Zoo (n = 9). The 12 zoos have adequate facilities to accommodate the different species of primates in indoor enclosure, different species live on separated places, and the feed managements are according to the Standard Rule of Chinese Association of Zoological Gardens. All animals' samples were collected by visiting once. At the time of faecal collections, there were no reported case of diarrhoea in the NHPs. Fresh feces were collected and packed in clear, self-sealing, disposable plastic bags marked with ID numbers, and transported in ice-filled foam boxes. Samples were stored in 2.5% potassium dichromate at 4˚C until DNA was extracted.

### DNA extraction and polymerase chain reaction (PCR)

Fecal samples were washed in distilled water to remove the potassium dichromate. Genomic DNA was then extracted using the PowerSoil® DNA Isolation Kit (MoBio, Carlsbad CA, USA), following manufacturer's instructions. DNA was stored at -20˚C prior to PCR analysis.

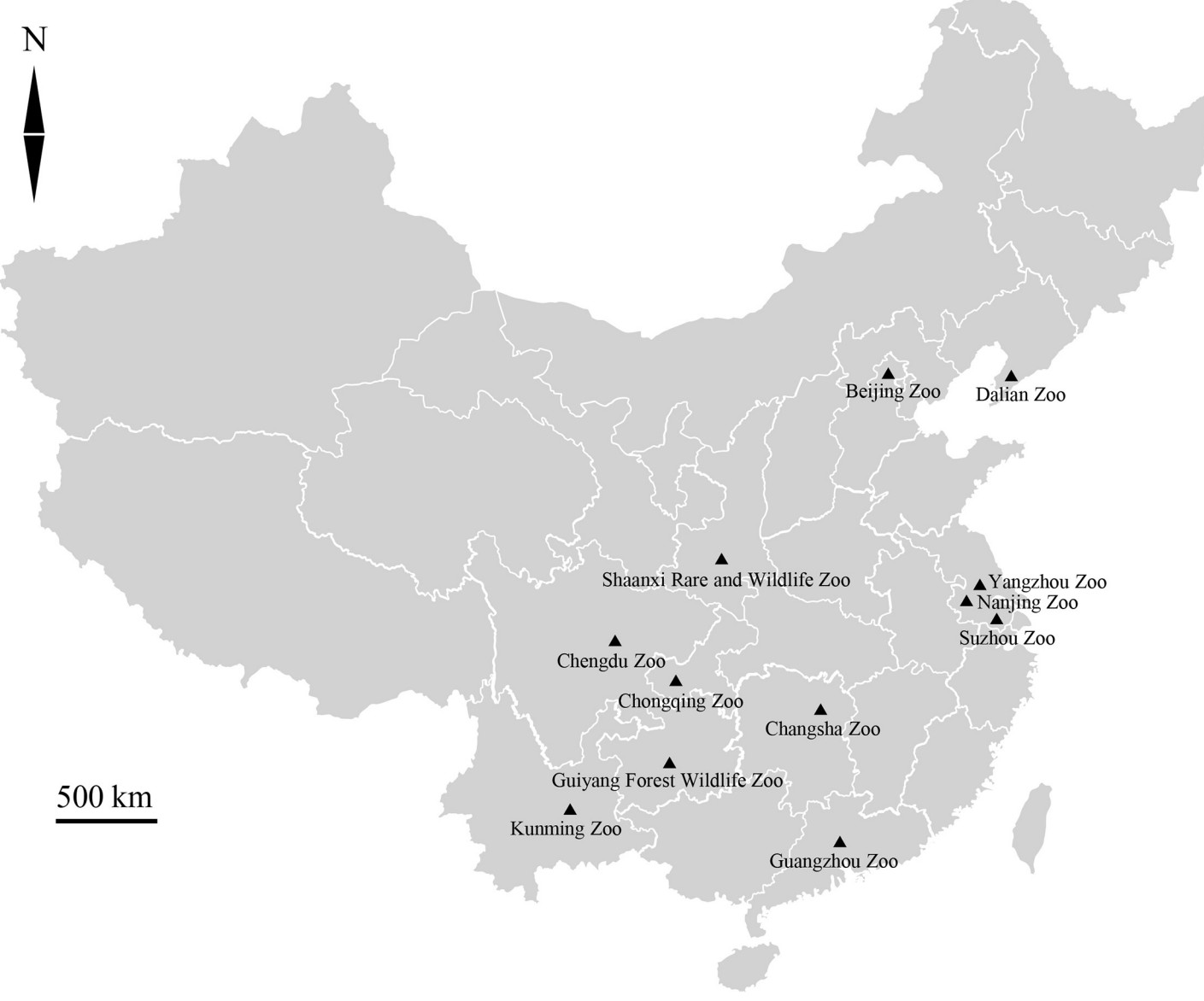

**Fig 1. Distribution of sampling sites from 12 zoos in China in this study.** Sampling sites are indicated by black triangles.

The PCR primers and protocol used in this study were previously described [15]. The PCR reactions for the *bg*, *tpi* and *gdh* loci were conducted in 25 μL reaction mixtures containing of 12.5 μL 2× Taq PCR Master Mix (KT201-02, Tiangen, Beijing, China), 8.5 μL deionized water (Tiangen, Beijing, China), 2 μL DNA, and 1 μL each of set primers, respectively. The primers and annealing temperatures for the three genes were listed in Table 1. Secondary PCR products were visualized by 1% agarose gel electrophoresis and staining with Golden View.

## Sequence analysis

All positive secondary PCR products were sequenced by BGI Tech Solutions (Liuhe Beijing) Co., Limited and were sequenced in both directions. Sequences were aligned with reference sequences from the GenBank database using BLAST (http://blast.ncbi.nlm.nih.gov) and

**Table 1. Primer sequences, annealing temperatures and the fragment lengths of the genes used in this study.**

| Gene | Primers | Sequence(5'-3') | Annealing Temperature(°C) | Fragment Length(bp) |
|------|---------|-----------------|---------------------------|---------------------|
| *bg* | F1 | AAGCCCGACGACCTCACCCGCAGTGC | 60 | 530 |
|      | R1 | GAGGCCGCCCTGGATCTTCGAGACGAC | | |
|      | F2 | GAACGAACGAGATCGAGGTCCG | 55 | |
|      | R2 | CTCGACGAGCTTCGTGTT | | |
| *tpi* | F1 | AAATIATGCCTGCTCGTCG | 50 | 530 |
|      | R1 | CAAACCTTITCCGCAAACC | | |
|      | F2 | CCCTTCATCGGIGGTAACTT | 50 | |
|      | R2 | GTGGCCACCACICCCGTGCC | | |
| *gdh* | F1 | TTCCGTRTYCAGTACAACTC | 50 | 511 |
|      | R1 | ACCTCGTTCTGRGTGGCGCA | | |
|      | F2 | ATGACYGAGCTYCAGAGGCACGT | 50 | |
|      | R2 | GTGGCGCARGGCATGATGCA | | |

Clustal X (http://www.clustal.org/). To evaluate the MLGs of *G. duodenalis*, we only included specimens that were successfully subtyped at all three loci and sequences with ambiguous positions (double peaks) were not included for phylogenetic analyses. Sequences were concatenated for each positive isolate to form a multilocus sequence (*bg* + *tpi* + *gdh*). All concatenated MLGs were used in a neighbour-joining analysis using the Kimura-2 parameter model calculated with MEGA 7 (http://www.megasoftware.net/). Representative nucleotide sequences obtained in this study were deposited in GenBank under the accession numbers: MK909127, MK909131, MK909135, MK909136, MK952610, MK952598, and MK952606.

## Results and discussion

In this study, infected NHPs were detected from six zoos of the 12 examined zoos. Suzhou Zoo had the highest infection rate (40.0%, 4/10), followed by Yangzhou Zoo (22.2%, 2/9), Dalian Zoo (16.7%, 4/24), Chengdu Zoo (12.8%, 6/47), Guiyang Forest Wildlife Zoo (12.1%, 7/58), and Changsha Zoo (4.7%, 2/43) (Table 2). The infection rate in Suzhou Zoo was closed to Changsha Wild Animal Zoo (44.0%, 33/75) [8]. Yangzhou Zoo and Dalian Zoo were closed to our previous study from Guiyang Zoo (30.0%, 15/50) [15]. Chengdu Zoo and Guiyang Forest Wildlife Zoo were closed to a public park in Guiyang (8.5%, 35/411) [20]. The infection rate in Changsha Zoo was similar to that detected in Wuhan Zoo (7.6%, 5/66) [8] and Guangxi Zoo (2.4%, 5/205) [21]. The various infection rates in different zoos may relate to geographic distribution [8, 11, 12, 14–16].

Twenty-five samples were positive for *G. duodenalis*, based on positive PCR results at any of the three genetic loci (*bg*, *tpi*, and *gdh*). Average infection rate in this study was 8.3% (25/302), which was lower than a previous study in captive NHPs from seven zoos (18.6%, 92/496) (Shijiazhuang Zoo, Wuhan Zoo, Taiyuan Zoo, Changsha Wild Animal Zoo, Beijing Zoo, Shanghai Zoo and Shanghai Wild Animal Zoo) and also lower than our previous study from three zoos (17.7%, 15/85) in southwestern China (Guiyang Zoo, Bifengxia Zoo and Chengdu Zoo) [8, 15]. Compared with other countries, the average infection rate in this study was closed to that in Thailand (7.0%, 14/200) [9] and Uganda (11.1%, 9/81) [10], but lower than that in North-West India (31.2%, 53/170) [11]. The differences of infection rates in NHPs may be related to animal health status, detection methods, or geo-ecological conditions [2, 8, 15, 16, 22–29]. Nested PCR protocols based on single-copy genes (*bg*, *tpi* and *gdh*) had considerable lower diagnostic sensitivities than those based on multiple-copy genes (e.g. *SSU rRNA*). In this

**Table 2. Occurrence and assemblage B of *G. duodenalis* for NHPs from 12 zoos in China.**

| Zoos name | Province | Positive NHPs species (n) | No. tested | No.(%)of positive specimens | 95% CI | Assemblage (n) |
|---|---|---|---|---|---|---|
| Beijing Zoo | Beijing* | – | 12 | 0 (0) | – | – |
| Chengdu Zoo | Sichuan | Golden monkey (6) | 47 | 6 (12.8%) | [2.9, 22.7] | B (6) |
| Changsha Zoo | Hunan | Ring-tailed lemur (2) | 43 | 2 (4.7%) | [-1.9, 11.2] | B (2) |
| Chongqing Zoo | Chongqing* | – | 33 | 0 (0) | – | – |
| Dalian Zoo | Liaoning | Chimpanzee (2) | 24 | 4 (16.7%) | [0.6, 32.7] | B (4) |
| | | Golden monkey (1) | | | | |
| | | Ring-tailed lemur (1) | | | | |
| Guiyang Forest Wildlife Zoo | Guizhou | Golden monkey (5) | 58 | 7 (12.1%) | [3.4, 20.7] | B (7) |
| | | Baboons (1) | | | | |
| | | White-cheeked gibbon (1) | | | | |
| Guangzhou Zoo | Guangdong | – | 8 | 0 (0) | – | – |
| Kunming Zoo | Yunnan | – | 16 | 0 (0) | – | – |
| Nanjing Zoo | Jiangsu | – | 16 | 0 (0) | – | – |
| Shaanxi Rare and Wildlife Zoo | Shaanxi | – | 26 | 0 (0) | – | – |
| Suzhou Zoo | Jiangsu | Ring-tailed lemur (1) | 10 | 4 (40.0%) | [3.1, 76.9] | B (4) |
| | | Japanese macaque (1) | | | | |
| | | Ruffed lemur (1) | | | | |
| | | Africa black-and-white colobus(1) | | | | |
| Yangzhou Zoo | Jiangsu | Squirrel monkey (1) | 9 | 2 (22.2%) | [-11.7, 56.1] | B (2) |
| | | Ring-tailed lemur (1) | | | | |
| Total: 12 zoos | Eight provinces and two municipalities | Golden monkey (12) | 302 | 25 (8.3%) | [5.2, 11.4] | B (25) |
| | | Squirrel monkey (1) | | | | |
| | | Japanese macaques (1) | | | | |
| | | Baboons (1) | | | | |
| | | Africa black-and-white colobus(1) | | | | |
| | | White-cheeked gibbon (1) | | | | |
| | | Chimpanzee (2) | | | | |
| | | Ring-tailed lemur (5) | | | | |
| | | Ruffed lemur (1) | | | | |

"*": municipality

study, we adopted single-copy genes (*bg*, *tpi* and *gdh*) for genotyping *G. duodenalis*, not use the multiple-copy gene (*SSU rRNA*), which may underestimate the true infection rates [19]. *Giardia duodenalis* infection in NHPs suggests more attention should be paid to the living conditions of NHPs, and a safe distance maintained between NHPs and humans [20].

For PCR analysis of 302 samples from 32 NHP species, only nine species were positive for *G. duodenalis*, including africa black-and-white colobus (100%, 1/1), ruffed lemur (50.00%, 1/2), ring-tailed lemur (31.25%, 5/16), japanese macaque (33.33%, 1/3), chimpanzee (22.22%, 2/9), golden monkey (17.39%, 12/69), white-cheeked gibbon (7.14%, 1/14), baboons (4.35%, 1/23) and squirrel monkey (3.33%, 1/30). The infection rates ranged from 3.33% to 100% in the nine NHPs species. The infection rates for chimpanzee in this study were higher than that reported in other studies [8, 15–17, 19]. Previous studies reported high infection rates in captive NHPs were concentrated on rhesus macaque (8.49%, 9/106), crab-eating macaque

(38.89%, 7/18), pig-tailed macaque (56.25%, 9/16), ring-tailed lemur (57.78%, 26/45), green monkey (20.00%, 3/15), hussar monkey (31.25%, 5/16), yellow baboon (40.00%, 2/5), cheeked gibbon (38.89%, 14/36), and bornean orangutan (21.74%, 5/23) [8, 15–17]. However, the rhesus macaque, crab-eating macaque, pig-tailed macaque, green monkey, mandrill, hussar monkey and Francois' leaf monkey were all found negative in our present study. The variation of infection rate for *G. duodenalis* in different NHPs species needs more studies to elucidate.

To date, assemblage A, B, and E have been identified in NHPs, with assemblage B dominating in China [8, 15–21]. In this study, all the *G. duodenalis*-positive specimens were assemblage B, which is consistent with previous studies [15, 16, 18]. Assemblage B is common in humans worldwide [6, 17, 27, 28]; therefore, NHPs may contribute to sporadic human infection [23, 24]. Of the 25 *G. duodenalis*-positive specimens, the *bg*, *tpi*, and *gdh* loci were successfully amplified and sequenced from 21, 21, and 20 specimens, respectively (Table 3). The *bg*, *tpi*, and *gdh* loci were highly polymorphic, with the greatest genetic variation at the *tpi* locus. Of the *bg* subtypes, three had previously been identified and the sequence of the remaining subtype BIII-1 (MK909127) was previously unpublished. Of the four *tpi* subtypes previously identified and the four remaining sequence subtypes, MB10-1 (MK909131), WB8-1 (MK909135), B14-1 (MK909136), and MB9-1(MK952610) were previously unpublished. Of the *gdh* subtypes, four were known and two had not been published (BIV-1 [MK952606] and DN7-1 [MK952598]). Twelve single nucleotide polymorphisms (SNPs) were detected within assemblage B at the *bg* / *tpi* / *gdh* loci (S2 Table). At the *bg* locus, two SNPs detected in isolate DLGB04. At the *tpi* locus, eight SNPs detected in six isolates (DLGT04, GYGT23, GYGT26, GYGT28, GYGT55 and SZGT10). At the *gdh* locus, two SNPs detected in three isolates (YZGG05, YZGG06 and GYGG97). Extensive polymorphism at the *bg*, *tpi*, and *gdh* loci in this study may reflect the wide geographic distribution of fecal samples. Previous studies demonstrated more genetic variation at the *tpi* locus (11, 7, and 3 novel sub-assemblage, respectively) [16, 18, 21]; however, our previous study found more variation at the *bg* locus [15]. NHPs in seven zoos in China [8]and wild rhesus macaques in India [11] had more genetic variation at the *bg* locus. The reason for more genetic variations at the *bg* and *tpi* loci is not clear.

To better understand the diversity of *G. duodenalis* infection, we used multilocus genotyping. Seventeen isolates from NHPs yielded 12 MLGs (MLG-SW10 to MLG-SW21). The most common MLG was MLG-SW15 (17.65%, 3/17), followed by MLG-SW18 (11.76%, 2/17), MLG-SW19 (11.76%, 2/17), and MLG-SW21 (11.76%, 2/17). The remaining MLGs were only detected in one specimen. Moreover, seven MLGs (MLG-SW10-12, 15–18) were identified in golden monkeys and three types of MLGs (MLG-SW13, 14, 21) were identified in ring-tailed lemurs. More MLGs detected in golden monkeys and ring-tailed lemurs implies a higher relative genetic diversity [8].

A phylogenetic, evolutionary tree based on concatenated sequences was constructed to better understand the diversity and relationship between MLGs in NHPs and humans (Fig 2). Of the 12 MLGs we identified, 11 clustered with the NHPs isolates and 1 MLG (MLG21) clustered with human isolates from Sweden. The MLGs formed two main clades, MLG-SW (10–12, 18) and MLG-SW (13, 14, 16, 17). MLG-SW (15, 19, 20, 21) was scattered throughout the phylogenetic tree. The role of NHPs in the transmission of *G. duodenalis* to humans is not clear; however, the occurrence of assemblage B detected in captive NHPs suggests transmission from humans or an adaptation to primate host [4, 8].

## Conclusion

This study evaluated the occurrence of *G. duodenalis* in NHPs from 12 zoos distributed across 8 provinces and 2 municipalities (Chongqing and Beijing) in China. All *G. duodenalis*

**Table 3. Multi-locus sequences of *bg*, *tpi* and *gdh* genes for 25 *G. duodenalis* positive faecal samples.**

| Geographic source (China) | Isolate | Host | Subtype / Host or source / GenBank accession number | | | MLGs |
|---|---|---|---|---|---|---|
| | | | *β-giardin* | *tpi* | *gdh* | |
| Chengdu Zoo | CDZOO36 | Golden monkey | Bb-1/squirrel monkey/ KJ888974 | – | – | – |
| | CDZOO38 | Golden monkey | – | B14/rhesus macaque/ KF679737 | – | – |
| | CDZOO39 | Golden monkey | Bb-1/squirrel monkey/ KJ888974 | B14/rhesus macaque/ KF679737 | BIV/rhesus macaque/ KF679731 | SW10[#] |
| | CDZOO40 | Golden monkey | Bb-1/squirrel monkey/ KJ888974 | B14/rhesus macaque/ KF679737 | BIV/japanese macaque/ KF679730 | SW11[#] |
| | CDZOO42 | Golden monkey | Bb-1/squirrel monkey/ KJ888974 | – | – | – |
| | CDZOO47 | Golden monkey | Bb-1/squirrel monkey/ KJ888974 | B14/rhesus macaque/ KF679737 | B:DN2/Homo sapiens/ MG746605 | SW12[#] |
| Changsha Zoo | CSZOO23 | Ring-tailed lemur | Bb-4/ring-tailed lemur/ KJ888977 | BIV/Homo sapiens/ HG970113 | BIV/japanese macaque/ KF679730 | SW13[#] |
| | CSZOO41 | Ring-tailed lemur | Bb-4/ring-tailed lemur/ KJ888977 | BIV/Homo sapiens/ HG970113 | Bh-2/ring-tailed lemur/ KJ888982 | SW14[#] |
| Dalian Zoo | DLZOO1/ DLZOO8 | Chimpanzee | B3 like4/Homo sapiens/ KT948089 | MB2/rhesus macaque/ KF679740 | Bh-1/squirrel monkey/ KJ888981 | SW15 |
| | DLZOO2 | Golden monkey | B3 like4/Homo sapiens/ KT948089 | MB2/rhesus macaque/ KF679740 | Bh-1/squirrel monkey/ KJ888981 | SW15 |
| | DLZOO4 | Ring-tailed lemur | BIII-1/lemur catta/ MK909127[#] | MB10-1/lemur catta/ MK909131[#] | – | – |
| Guiyang Forest Wildlife Zoo | GYZOO23 | Golden monkey | B/Homo sapiens/ FJ560593 | WB8-1/golden monkey/ MK909135[#] | BIV/rhesus macaque/ KF679731 | SW16[#] |
| | GYZOO24 | Golden monkey | B3 like4/Homo sapiens/ KT948089 | BIV/Homo sapiens/ HG970113 | BIV/rhesus macaque/ KF679731 | SW17[#] |
| | GYZOO25 | Golden monkey | – | – | B/Homo sapiens/KT948096 | – |
| | GYZOO26/ GYZOO28 | Golden monkey | Bb-1/squirrel monkey/ KJ888974 | B14-1/golden monkey/ MK909136[#] | BIV/japanese macaque/ KF679730 | SW18[#] |
| | GYZOO55 | Baboons | B/Homo sapiens/ FJ560593 | WB8-1/baboons/ MK909135[#] | – | – |
| | GYZOO97 | White-cheeked gibbon | – | – | BIV-1/rhesus macaque/ MK952606[#] | – |
| Suzhou Zoo | SZZOO3 | Japanese macaque | B3 like4/Homo sapiens/ KT948089 | BIV/Homo sapiens/ HG970113 | BIV/rhesus macaque/ KF679729 | SW19[#] |
| | SZZOO5 | Ruffed Lemur | B3 like4/Homo sapiens/ KT948089 | BIV/Homo sapiens/ HG970113 | BIV/rhesus macaque/ KF679729 | SW19[#] |
| | SZZOO9 | Africa Black-and-white Colobus | B3 like4/Homo sapiens/ KT948089 | MB9/ring-tailed lemur/ KJ888985 | BIV/japanese macaque/ KF679730 | SW20[#] |
| | SZZOO10 | Ring-tailed lemur | – | MB9-1/ring-tailed lemur/ MK952610[#] | Bh-2/ring-tailed lemur/ KJ888982 | – |
| Yangzhou Zoo | YZZOO5 | Squirrel monkey | B3 like4/Homo sapiens/ KT948089 | BIV/Homo sapiens/ HG970113 | DN7-1/squirrel monkey/ MK952598[#] | SW21[#] |
| | YZZOO6 | Ring-tailed lemur | B3 like4/Homo sapiens/ KT948089 | BIV/Homo sapiens/ HG970113 | DN7-1/ring-tailed lemur/ MK952598[#] | SW21[#] |

"#": Novel subtypes and novel MLGs: "MLG-SW" follow by our previous study [15].

infections belonged to assemblage B, including seven novel subtypes: BIII-1, MB10-1, WB8-1, B14-1, MB9-1, DN7-1, and BIV-1. The *tpi* locus was the most genetically heterogeneous of the three loci evaluated. Multilocus genotyping identified twelve different assemblage B MLGs (one known MLG and eleven novel MLGs), implied relative higher genetic diversity. This

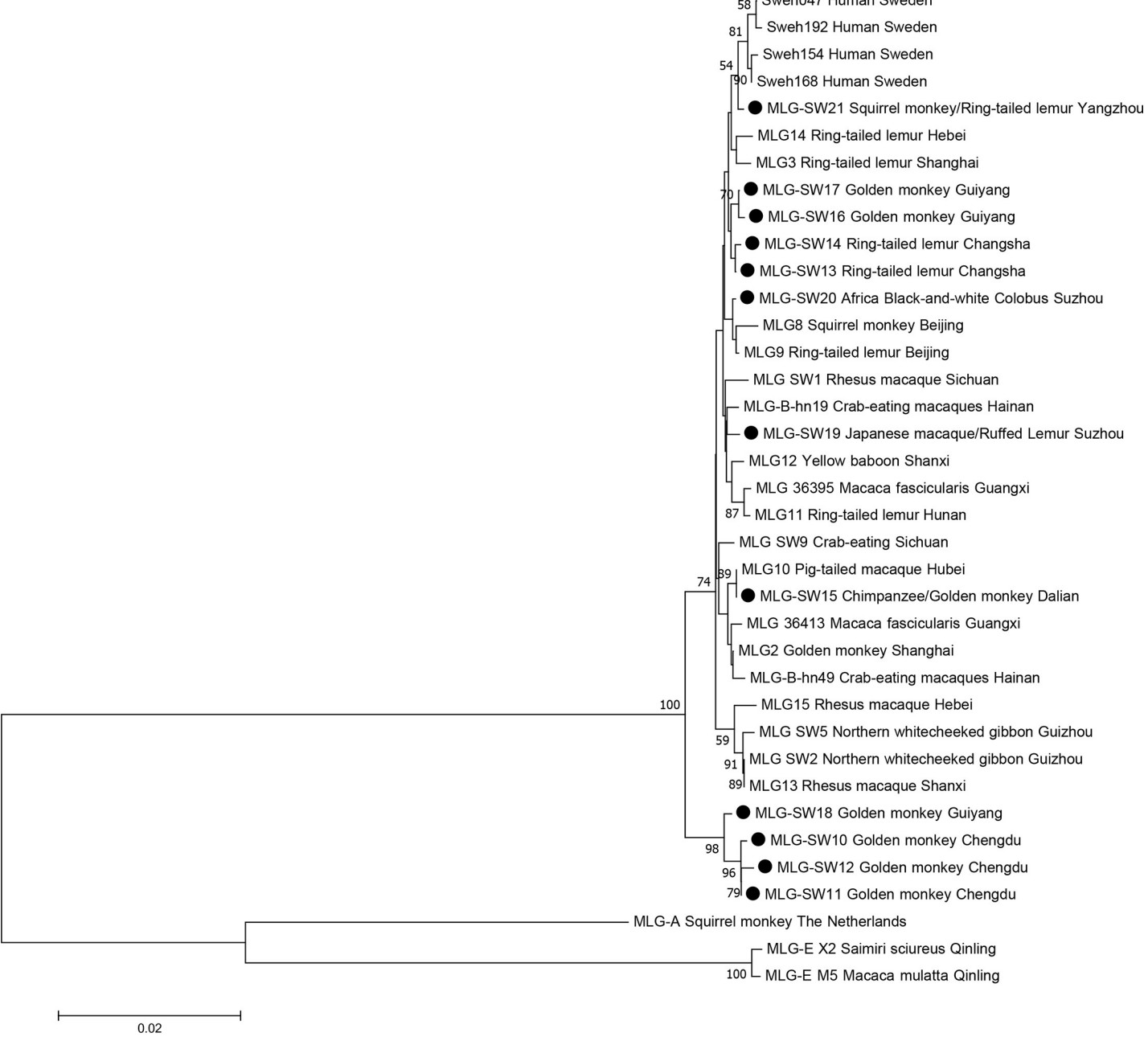

**Fig 2. Phylogenetic relationship of *G. duodenalis* assemblage B multilocus genotypes (MLGs) inferred by the neighbor-joining analysis of concatenated *bg*, *tpi*, and *gdh* sequences.** Reference sequences used are from the studies by Karim et al.[8], Levecke et al.[12], Zhong et al.[15], Chen et al.[18], Ye et al.[21], Lebbadet al.[28] and Du et al.[30]. Bootstrap values greater than 50% from 1000 replicates are shown. Concatenated sequences from this study are marked by filed roundness.

study enlarge our understanding using multilocus genotyping of *Giardia* infection for captive NHPs from 12 zoos in China. Further research on the potential spread of NHPs *G. duodenalis* to humans needed more data to elucidate.

## Supporting information

**S1 Table. Occurrence of *Giardia duodenalis* in different species of nonhuman primates.**
(DOC)

**S2 Table. Variations in *bg*, *tpi* and *gdh* nucleotide sequences among the subtypes of *Giardia duodenalis* assemblage B from NHPs.**
(DOCX)

# Acknowledgments

We thank Liqin Wang and the zoo staff for their assistance in sample collection during this study.

# Author Contributions

**Conceptualization:** Guangneng Peng, Yufei Wang, Zhijun Zhong.

**Data curation:** Hualin Fu, Xiaobin Gu.

**Formal analysis:** Xueping Zhang, Liuhong Shen, Xiaoping Ma, Haifeng Liu.

**Investigation:** Xueping Zhang, Shumin Yu, Changliang He.

**Methodology:** Xueping Zhang, Jiaming Dan.

**Resources:** Liqin Wang, Suizhong Cao.

**Software:** Ya Wang, Ziyao Zhou.

**Supervision:** Guangneng Peng, Zhijun Zhong.

**Validation:** Liqin Wang, Xinting Lan, Zhihua Ren, Zhicai Zuo.

**Writing – original draft:** Xinting Lan, Jiaming Dan.

**Writing – review & editing:** Junliang Deng, Yanchun Hu, Yi Geng.

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
