## [Decision Letter · Decision Letter 0]

10 Oct 2019

PONE-D-19-24653

Infection prevalence and multilocus genotyping of Giardia duodenalis in captive non-human primates from 12 zoos in China

PLOS ONE

Dear Dr. Zhong,

Thank you for submitting your manuscript to PLOS ONE. After careful consideration, we feel that it has merit but does not fully meet PLOS ONE’s publication criteria as it currently stands. Therefore, we invite you to submit a revised version of the manuscript that addresses the points raised during the review process.

We would appreciate receiving your revised manuscript by Nov 24 2019 11:59PM. To enhance the reproducibility of your results, we recommend that if applicable you deposit your laboratory protocols in protocols.io, where a protocol can be assigned its own identifier (DOI) such that it can be cited independently in the future. For instructions see: http://journals.plos.org/plosone/s/submission-guidelines#loc-laboratory-protocols

We look forward to receiving your revised manuscript.

Kind regards,

Hesham

Hesham M. Al-Mekhlafi, PhD

Academic Editor

PLOS ONE

Journal Requirements:

Additional Editor Comments (if provided):

Reviewers' comments:

Reviewer's Responses to Questions

**Comments to the Author**

1. Is the manuscript technically sound, and do the data support the conclusions?

Reviewer #1: Yes

Reviewer #2: Yes

2. Has the statistical analysis been performed appropriately and rigorously? 

Reviewer #1: N/A

Reviewer #2: No

3. Have the authors made all data underlying the findings in their manuscript fully available?

Reviewer #1: Yes

Reviewer #2: Yes

4. Is the manuscript presented in an intelligible fashion and written in standard English?

Reviewer #1: Yes

Reviewer #2: No

5. Review Comments to the Author

Reviewer #1: In this manuscript Zhang et al report the molecular epidemiology of the diarrhoea-causing enteric protozoan parasite Giardia duodenalis infecting captive non-human primates in 12 zoological Gardens in China. Detection of the parasite was conducted from faecal samples by PCR methods targeting partial fragments of the gdh, bg, and tpi loci. This multilocus genotyping approach allowed the identification of novel genetic variants of the parasite. Overall, this study has certain epidemiological relevance, although there are a number of issues that need addressing (see below).

Major issues

1. Detection of the parasite was based on single-copy genes. These genes are particularly suited for genotyping analyses, but their use as diagnostic tools is hampered by limited sensitivity. For this particular purpose a PCR method targeting the multiple-copy ssu rRNA locus of the parasite would yield a higher number of positive results. In practical terms this means that the infection rates reported here are very likely an underestimation of the true figures. This issue should me mentioned as a limitation of the study in the Discussion section.

2. The term ´prevalence´ is probably not appropriate in this survey, as in 7/12 zoos only24 or less faecal samples were collected. I would suggest replacing the term by infection rate or occurrence rate. Same comment for the title of the paper.

3. Abstract section, lines 42-43: please provide more information about how the nomenclature used to name MLGs were chosen. For instance, what is the meaning of SW? It seems that the nomenclature used here is quite arbitrary. Please clarify.

4. Introduction section: please provide more information regarding the current molecular epidemiological situation of G. duodenalis infections in human and non-human primates in China. Briefly describe the range of prevalence rates reported in the literature, the diversity and frequency of assemblages/sub-assemblages identified, and any relevant differences between geographical locations and, if available, between captive and free-living NHP. Mention also if there is information regarding potential zoonotic transmission events between NHP and HP (or vice versa). This information would help the interested read to have a better picture of the current status of the infection in China.

5. M&M section: the Sample collection sub-section is poorly described. Please clearly state how zoological gardens were selected, approached and invited to participate in the survey. Which were the inclusion/exclusion criteria for selecting samples? Did the investigated NHP present any clinical manifestation (e.g. diarrhoea) at the moment of sampling? Did the Authors investigate the occurrence of other enteric pathogens? Please develop.

6. M&M section: the description of the molecular methods used in the present study for the detection and genotyping of Giardia duodenalis are poorly described. Please note that enough information should be provided here to enable the interested reader to repeat the experiments without the need of checking primary sources. Please thoroughly describe primer sequences, reagent concentrations, cycling conditions, and equipment used. Indicate also the percentage of the agarose gels used during electrophoresis.

7. M&M section: regarding sequencing, please clarify whether sequencing was conducted in both directions or not. Also, did the Authors check for the presence of ambiguous positions (double peaks) during chromatogram inspection? SNPs are frequently reported at the three loci investigated in the present survey. Importantly, please confirm that only sequences without double peaks were used in the phylogenetic analyses, as the presence of ambiguous positions would bias the analysis. I was unable to check this point as provided GenBank accession numbers are not accessible yet.

8. Results section: data presented in the paper do not allow to identify the genetic diversity fount in the gdh, bg, and tpi sequences generated in the present study. I would recommend conducting multiple sequence alignment analysed with appropriate reference sequences to identify SNPs. These results can be shown as a new Figure, or summarized in a Table.

9. Table 2 only shows results for 23 NHP, not 25 as indicated in the legend and the main body of the manuscript. Also, isolates described at the bg locus are indicated as B3 or BIII. Are those the same? If necessary, please standardise the nomenclature to avoid confusion in naming assemblages and sub-assemblages.

Minor issues

1. Line 51: Giardia duodenalis (in full at the beginning of a sentence). Same comment for lines 53, 56, etc. Please amend.

2. Line 88: Amplicons of the expected size were…

3. Line 127: Giardia should be italicised.

Reviewer #2: The study expanded to wider area in China and included a greater number of zoos compared with the author’s previous study published in 2017. However, besides reporting of new genotypes, there are no in-depth analyses or good number of positive samples available that can contribute conclusive information that related to geographic segregation, host-adaptation and impacts on transmission.

1. Line 84-86: Please further explain automatic gel electrophoresis analysis by Giardia PCR.

2. The title of table 1 do not sound right, the presentation of the data should be improved.

Suggest combine location and species infection rate (S1 Table) to make it more comprehensive and include the species infection rate in the discussion. Eg. The high infection rate of ring-tailed lemur (31.25), it’s from a single or multiple zoos?

3. The prevalence of giardiasis ranged from 0 to 40%, any background information of the zoo eg. Environment, management to explain?

4. Line 108-115: The overall prevalence rate is 8.3%, again it’s ranged from 0-40% from different zoos, it’s not meaningful to compare this overall rate with single-centre/ single-location study from previous papers.

5. Since the samples were collected from different provinces, perhaps a map can be included to show where the samples were collected alongside with the phylogenetic tree.

6. PLOS authors have the option to publish the peer review history of their article (what does this mean?). If published, this will include your full peer review and any attached files.

Reviewer #1: Yes: David Carmena

Reviewer #2: No

---

## [Author Response · Author response to Decision Letter 0]

21 Nov 2019

Response to Reviewer:

Thank you to the reviewers for their time and thoughtful comments, many of which have been incorporated into the revised manuscript. This revision was carried out according to the suggestions of the reviewers. Detailed comments are as follows.

Reviewer #1:

Major

1. Detection of the parasite was based on single-copy genes. These genes are particularly suited for genotyping analyses, but their use as diagnostic tools is hampered by limited sensitivity. For this particular purpose a PCR method targeting the multiple-copy ssu rRNA locus of the parasite would yield a higher number of positive results. In practical terms this means that the infection rates reported here are very likely an underestimation of the true figures. This issue should me mentioned as a limitation of the study in the Discussion section. 

Answer: Thanks for your comments. We added some description about the limitation of our present method in the Discussion section and highlighted it in the revised tracked-manuscript. (lines 149 -151)

2. The term ´prevalence´ is probably not appropriate in this survey, as in 7/12 zoos only24 or less faecal samples were collected. I would suggest replacing the term by infection rate or occurrence rate. Same comment for the title of the paper.

Answer: Thanks for your comments. We modified the term ´prevalence´ into ´occurrence´and highlighted it in the revised manuscript. 

3. Abstract section, lines 42-43: please provide more information about how the nomenclature used to name MLGs were chosen. For instance, what is the meaning of SW? It seems that the nomenclature used here is quite arbitrary. Please clarify.

Answer: Thanks for your comments. The nomenclature used to name MLGs in the present study was consistent with our previous study [1] and the meaning of SW (southwest) is a number name which used in our laborary for the NHP samples. Our lab located in the southwest of China.(lines 212-213) 

4. Introduction section: please provide more information regarding the current molecular epidemiological situation of G. duodenalis infections in human and non-human primates in China. Briefly describe the range of prevalence rates reported in the literature, the diversity and frequency of assemblages/sub-assemblages identified, and any relevant differences between geographical locations and, if available, between captive and free-living NHP. Mention also if there is information regarding potential zoonotic transmission events between NHP and HP (or vice versa). This information would help the interested read to have a better picture of the current status of the infection in China.

Answer: Thanks for your comments. We added some description about the epidemiological situation of G. duodenalis infections in non-human primates, potential zoonotic transmission events between NHP and HP. All the changes were highlighted in the revised manuscript (line 59-71).

5. M&M section: the Sample collection sub-section is poorly described. Please clearly state how zoological gardens were selected, approached and invited to participate in the survey. Which were the inclusion/exclusion criteria for selecting samples? Did the investigated NHP present any clinical manifestation (e.g. diarrhoea) at the moment of sampling? Did the Authors investigate the occurrence of other enteric pathogens? Please develop.

Answer: Thanks for your comments. In this study, we initially selected zoos in each of the nine geographical regions (Central China, North China, East China, South China, West China, Northwestern China, Northeastern China, Southwest China, Southeastern China) of China, but only get 12 zoos permission to collect samples. The 12 zoos were distributed in seven different geographical divisions [Central China: Hunan province (Changsha Zoo); North China: Beijing (Beijing Zoo); East China: Jiangsu province ( Suzhou Zoo, Yangzhou Zoo and Nanjing Zoo); Southern China: Guangdong province (Guangzhou Zoo); Northwestern China: Shaanxi (Shaanxi Rare and Wildlife Zoo); Northeastern China: Liaoning province (Dalian Zoo); Southwest China: Sichuan province (Chengdu Zoo), Chongqing (Chongqing Zoo), Guizhou province (Guiyang Forest Wildlife Zoo), Yunnan province (Kunming Zoo)]. We added a map to better illustrate the location of the zoos in the revised manuscript (Fig 1) ( lines 94 -95). 

All animals sampled in this study were collected by visiting once. At the time of faecal collections, there were no reported cases of diarrhoea in the zoos. In present study, we only focus on the occurrence of G. duodenalis infections in non-human primates. We added relevant description and highlighted it in the revised manuscript( lines 85-89).

6. M&M section: the description of the molecular methods used in the present study for the detection and genotyping of Giardia duodenalis are poorly described. Please note that enough information should be provided here to enable the interested reader to repeat the experiments without the need of checking primary sources. Please thoroughly describe primer sequences, reagent concentrations, cycling conditions, and equipment used. Indicate also the percentage of the agarose gels used during electrophoresis.

Answer: Thanks for your comments. We added relevant description and Table 1 (including primer sequences, annealing temperatures and the fragment lengths of the genes used in this study) to give the details for readers. All the changes were highlighted in the revised manuscript (line103-107 and line123-125 ).

7. M&M section: regarding sequencing, please clarify whether sequencing was conducted in both directions or not. Also, did the Authors check for the presence of ambiguous positions (double peaks) during chromatogram inspection? SNPs are frequently reported at the three loci investigated in the present survey. Importantly, please confirm that only sequences without double peaks were used in the phylogenetic analyses, as the presence of ambiguous positions would bias the analysis. I was unable to check this point as provided GenBank accession numbers are not accessible yet.

Answer: Thanks for your comments. The PCR products were sequenced in both directions and the information was involved in our revised manuscript (line 110-111). Sequences with ambiguous positions (double peaks) were not included in this study. 

8. Results section: data presented in the paper do not allow to identify the genetic diversity fount in the gdh, bg, and tpi sequences generated in the present study. I would recommend conducting multiple sequence alignment analysed with appropriate reference sequences to identify SNPs. These results can be shown as a new Figure, or summarized in a Table.

Answer: Thanks for your comments. We added a table about SNPs as a supplementary (S2 Table) and relevant description in the revised manuscript (line 188-193). 

9. Table 2 only shows results for 23 NHP, not 25 as indicated in the legend and the main body of the manuscript. Also, isolates described at the bg locus are indicated as B3 or BIII. Are those the same? If necessary, please standardise the nomenclature to avoid confusion in naming assemblages and sub-assemblages.

 Answer: Thanks for your comments. There were indeed 25 G. duodenalis-positive samples, including two chimpanzee (DLZOO1/DLZOO8) and two golden monkey (GYZOO26/GYZOO28). B3 and BIII are two different subtypes, from two references and named differently[2, 3].

Minor

1. Line 51: Giardia duodenalis (in full at the beginning of a sentence). Same comment for lines 53, 56, etc. Please amend.

Answer: Thanks for your comments. We corrected the mistake and highlighted it in the revised manuscript.

2. Line 88: Amplicons of the expected size were…

Answer: Thanks for your comments. Amplicons of the expected size of three genes (bg, tpi, gdh) were 530, 530 and 511. We added Table 1 which included this information in the revised manuscript.

3. Line 127: Giardia should be italicised.

Answer: Thanks for your comments. We corrected the mistake and highlighted it in the revised manuscript.

Reviewer #2: 

=========

1. Line 84-86: Please further explain automatic gel electrophoresis analysis by Giardia PCR.

Answer: Thanks for your comments. This was our statement error, we corrected the mistake and highlighted it in the revised manuscript.( lines 107 -108).

2. The title of table 1 do not sound right, the presentation of the data should be improved. Suggest combine location and species infection rate (S1 Table) to make it more comprehensive and include the species infection rate in the discussion. Eg. The high infection rate of ring-tailed lemur (31.25), it’s from a single or multiple zoos?

Answer: Thanks for your comments. We added the positive NHPs species (n) in table 1 (now it is table 2) and relevant discussion about infection rates of NHPs. We use Table 2 to show the difference infention rate in twelve Zoo, while S1 Table is main to show the results for 32 NHPs species infection, it’s convenient for readers to get more useful information for the 12 Zoos and 32 different NHP species infection. All the changes were highlighted.( lines 154-169).

3. The prevalence of giardiasis ranged from 0 to 40%, any background information of the zoo eg. Environment, management to explain?

 Answer: Thanks for your comments. We add the more information about the information of the 12 zoos (line 85-89). 

4. Line 108-115: The overall prevalence rate is 8.3%, again it’s ranged from 0-40% from different zoos, it’s not meaning to compare this overall rate with single-centre/ single-location study from previous papers.

 Answer: Thanks for your comments. We delect the content about comparing with single zoo, we rewrite this parts of the content about the details of occurrence rates of G. duodenalis infection (single vs single, overall rate vs overall rate) ( lines 127-147).

5. Since the samples were collected from different provinces, perhaps a map can be included to show where the samples were collected alongside with the phylogenetic tree.

Answer: Thanks for your comments. We added a map (Fig 1) and related contents which were highlighted in the revised manuscrip ( lines 86, and 94-95).

References

1.Zhong Z, Tian Y, Li W, Huang X, Deng L, Cao S, et al. Multilocus genotyping of Giardia duodenalis in captive non-human primates in Sichuan and Guizhou provinces, Southwestern China. PLoS One. 2017;12(9):e0184913. doi: 10.1371/journal.pone.0184913. PMID: 28910395.

2.Coronato Nunes B, Pavan MG, Jaeger LH, Monteiro KJ, Xavier SC, Monteiro FA, et al. Spatial and Molecular Epidemiology of Giardia intestinalis Deep in the Amazon, Brazil. PLoS One. 2016;11(7):e0158805. Epub 2016/07/09. doi: 10.1371/journal.pone.0158805.PMID: 27392098.Wegayehu, T.

3.Karim, M. R. Li, J. Adamu, H. Erko, B. Zhang, L. Tilahun, G, et al. Multilocus genotyping of Giardia duodenalis isolates from children in Oromia Special Zone, central Ethiopia. BMC Microbiol. 2016. 16(1).doi: 10.1186/s12866-016-0706-7. PMID: 27209324.

---

## [Decision Letter · Decision Letter 1]

7 Jan 2020

PONE-D-19-24653R1

Occurrence and multilocus genotyping of Giardia duodenalis in captive non-human primates from 12 zoos in China

PLOS ONE

Dear Dr. Zhong,

Thank you for submitting your manuscript to PLOS ONE. After careful consideration, we feel that it has merit but does not fully meet PLOS ONE’s publication criteria as it currently stands. Therefore, we invite you to submit a revised version of the manuscript that addresses the points raised during the review process.

We would appreciate receiving your revised manuscript by Feb 21 2020 11:59PM. To enhance the reproducibility of your results, we recommend that if applicable you deposit your laboratory protocols in protocols.io, where a protocol can be assigned its own identifier (DOI) such that it can be cited independently in the future. For instructions see: http://journals.plos.org/plosone/s/submission-guidelines#loc-laboratory-protocols

We look forward to receiving your revised manuscript.

Kind regards,

Hesham

Hesham M. Al-Mekhlafi, PhD

Academic Editor

PLOS ONE

Reviewers' comments:

Reviewer's Responses to Questions

**Comments to the Author**

1. If the authors have adequately addressed your comments raised in a previous round of review and you feel that this manuscript is now acceptable for publication, you may indicate that here to bypass the “Comments to the Author” section, enter your conflict of interest statement in the “Confidential to Editor” section, and submit your "Accept" recommendation.

Reviewer #1: (No Response)

2. Is the manuscript technically sound, and do the data support the conclusions?

Reviewer #1: Yes

3. Has the statistical analysis been performed appropriately and rigorously? 

Reviewer #1: N/A

4. Have the authors made all data underlying the findings in their manuscript fully available?

Reviewer #1: Yes

5. Is the manuscript presented in an intelligible fashion and written in standard English?

Reviewer #1: Yes

6. Review Comments to the Author

Reviewer #1:

Lines 78-91: In my initial appraisal, I requested information about how zoological gardens were selected, approached, and invited to participate in the study. I also requested information regarding the inclusion/exclusion criteria for selecting samples, and the presence/absence of other parasitic, viral, and bacterial infections. None of these have been addressed in the revised version of the manuscript. Amend.

Lines 149-151: please note that this statement does not fully clarify the initial issue. Please clearly state here that PCR protocols based on single-copy genes (e.g. gdh, bg, and tpi) had considerable lower diagnostic sensitivities than those based on multiple-copy genes (e.g. ssu). In practical terms, this means that infection rates reported in the present study are an underestimation of the true ones. Amend.

Sequence analyses: please clearly state in the text that sequences with ambiguous positions (double peaks) were not included in the phylogenetic analyses.

7. PLOS authors have the option to publish the peer review history of their article (what does this mean?). If published, this will include your full peer review and any attached files.

Reviewer #1: Yes: David Carmena

---

## [Author Response · Author response to Decision Letter 1]

13 Jan 2020

Response to Reviewer:

Thank you to the reviewers for their time and thoughtful comments, many of which have been incorporated into the revised manuscript. This revision was carried out according to the suggestions of the reviewers. Detailed comments are as follows.

Reviewer #1:

Major

1.Lines 78-91: In my initial appraisal, I requested information about how zoological gardens were selected, approached, and invited to participate in the study. I also requested information regarding the inclusion/exclusion criteria for selecting samples, and the presence/absence of other parasitic, viral, and bacterial infections. None of these have been addressed in the revised version of the manuscript. Amend.

Answer: Thanks for your comments. Some related description are add in both Intrduction and Sample collection section, and highlighted them in the revised tracked-manuscript. (line 74-81, line 90-92 and line 95-98)

To date, most Chinese studies evaluating G. duodenalis infection in NHPs have focused on a single zoo or localized area. In this project, ongoing epidemiological surveys on intestinal zoonotic parasites of G. duodenalis, expanded previous studies to large-scale investigation of zoos and NHP species in China. As our last replied in the MS R1 version, in this study we only get 12 zoos permission to collect samples. The 12 zoos were distributed in seven different geographical divisions [Central China: Hunan province (Changsha Zoo); North China: Beijing (Beijing Zoo); East China: Jiangsu province ( Suzhou Zoo, Yangzhou Zoo and Nanjing Zoo); Southern China: Guangdong province (Guangzhou Zoo); Northwestern China: Shaanxi (Shaanxi Rare and Wildlife Zoo); Northeastern China: Liaoning province (Dalian Zoo); Southwest China: Sichuan province (Chengdu Zoo), Chongqing (Chongqing Zoo), Guizhou province (Guiyang Forest Wildlife Zoo), Yunnan province (Kunming Zoo)]. Our present project evaluated 302 NHP fecal samples (including 32 primate species) from 12 zoos distributed across eight Chinese provinces and two municipalities (Chongqing and Beijing), to better understand G. duodenalis infection in captive NHPs throughout China. In this project, we only focus on the occurrence of G. duodenalis infections in non-human primates, the presence/absence of other parasitic, viral, and bacterial infections is unknown. The 12 zoos have adequate facilities to accommodate the different species of primates in indoor enclosure, different species live on separated places. The 12 zoos are distributed throughout China (Fig 1), and the feed managements are all according to the Standard Rule of Chinese Association of Zoological Gardens. 

2.Lines 149-151: please note that this statement does not fully clarify the initial issue. Please clearly state here that PCR protocols based on single-copy genes (e.g. gdh, bg, and tpi) had considerable lower diagnostic sensitivities than those based on multiple-copy genes (e.g. ssu). In practical terms, this means that infection rates reported in the present study are an underestimation of the true ones. Amend.

Answer: Thanks for your comments. We added relevant description and highlighted it in the revised manuscript. (lines 162-164, lines 166-167) 

3.Sequence analyses: please clearly state in the text that sequences with ambiguous positions (double peaks) were not included in the phylogenetic analyses.

Answer: Thanks for your comments. We added relevant description and highlighted it in the revised manuscript. (lines 127-128)

---

## [Editor Report · Decision Letter 2]

22 Jan 2020

Occurrence and multilocus genotyping of Giardia duodenalis in captive non-human primates from 12 zoos in China

PONE-D-19-24653R2

Dear Dr. Zhong,

We are pleased to inform you that your manuscript has been judged scientifically suitable for publication and will be formally accepted for publication once it complies with all outstanding technical requirements.

With kind regards,

Hesham

Hesham M. Al-Mekhlafi, PhD

Academic Editor

PLOS ONE
---

## [Editor Report · Acceptance letter]

24 Jan 2020

PONE-D-19-24653R2 

Occurrence and multilocus genotyping of *Giardia duodenalis* in captive non-human primates from 12 zoos in China 

Dear Dr. Zhong:

I am pleased to inform you that your manuscript has been deemed suitable for publication in PLOS ONE. Congratulations! Your manuscript is now with our production department. 

With kind regards,

on behalf of

Dr. Hesham M. Al-Mekhlafi 

Academic Editor

PLOS ONE